# A Transparent Hydrogel-Ionic Conductor with High Water Retention and Self-Healing Ability

**DOI:** 10.3390/ma17020288

**Published:** 2024-01-06

**Authors:** Yangwoo Lee, Ju-Hee So, Hyung-Jun Koo

**Affiliations:** 1Department of Chemical & Biomolecular Engineering, Seoul National University of Science & Technology, 232 Gongneung-ro, Nowon-gu, Seoul 01811, Republic of Korea; diddnqkqh@gmail.com; 2Material & Component Convergence R&D Department, Korea Institute of Industrial Technology, Ansan 15588, Republic of Korea

**Keywords:** ionic gel, hydrogel, agarose, sensor

## Abstract

This study presents a transparent and ion-conductive hydrogel with suppressed water loss. The hydrogel comprises agarose polymer doped with sucrose and sodium chloride salt (NaCl–Suc/A hydrogel). Sucrose increases the water retention of the agarose gel, and the Na and Cl ions dissolved in the gel provide ionic conductivity. The NaCl–Suc/A gel shows high retention capability and maintains a 45% water uptake after 4 h of drying at 60 °C without encapsulation at the optimum gel composition. The doped NaCl–Suc/A hydrogel demonstrates improved mechanical properties and ionic conductivity of 1.6 × 10^−2^ (S/cm) compared to the pristine agarose hydrogel. The self-healing property of the gel restores the electrical continuity when reassembled after cutting. Finally, to demonstrate a potential application of the ion-conductive hydrogel, a transparent and flexible pressure sensor is fabricated using the NaCl–Suc/A hydrogel, and its performance is demonstrated. The results of this study could contribute to solving problems with hydrogel-based devices such as rapid dehydration and poor mechanical properties.

## 1. Introduction

Recently, interest in devices operated by ionic currents or ionics has considerably increased [1,2,3]. Since various processes in biological systems, such as adenosine triphosphate (ATP) synthesis [4,5,6], neural transmission [7,8,9], and cell-to-cell communication [10,11,12], are driven by ionic current, ionics can form a seamless bio-interface without the need for electronic-to-ionic current conversion [13], in contrast to conventional devices operated by electronic currents. Due to these advantages, the concept of ionics is attracting much attention, especially for wearable and implantable devices. For example, ionic communication-based neural interface devices have been introduced to use ions to noninvasively transmit neurophysiologic data from rodents [14]. Textile-based transistors that are composed of a gel with silica nanoparticles and ionic liquid are also reported as multiplexed diagnostic devices [15]. Recently, a review summarized and compared different types of ionic conductors for flexible and wearable triboelectric nano-generators [16]. As a matrix material for ionics, hydrogel is most actively used, owing to its mechanical properties of softness, flexibility and stretchability, high ion conductivity, and biocompatibility [16,17,18,19]. Also, hydrogels composed of non-covalent interactions typically exhibit self-healing properties, due to an ability to restore broken crosslinks [20,21,22]. Furthermore, ionics based on a hydrogel matrix can almost completely recover their ionic conductivity through simple physical re-contact, even if they are broken, and this feature can also be considered to be a self-healing property in a broad sense. Hydrogel has a structure where hydrophilic polymer chain networks hold a large amount of water [19]. Agarose gel is one of the hydrogels derived from hydrophilic biopolymers. Agarose polymer comprises the linearly arranged polysaccharide galactan and is produced via separating and purifying it from agar or seaweed containing agar components. Agarose-based hydrogels are biocompatible and exhibit thermally reversible sol–gel transition, allowing them to be easily molded into various shapes. As agarose polymers can hold water up to ~100 times their weight, gels with a very high water content can be prepared, resulting in high ionic conductivity, similar to that in a liquid environment [18]. These advantages make agarose-based hydrogel an appealing candidate as a matrix material for ionics. However, agarose-based hydrogels with high water content have relatively poor mechanical properties and, like other hydrogels, lose water and shrink rapidly over time, which needs to be resolved.

To enhance the water retention of hydrogel ionic conductors, there have been different approaches. Adding salts to hydrogels is the most common way to increase water retention due to the ionic solvation effect. Adding lithium chloride (LiCl) to polyacrylamide hydrogel showed 70% water retention after 5 days at very low humidity conditions (10% RH) [23]. Poly(sulfobetaine methacrylate(SBMA)-co-acrylic acid(AA)) hydrogel also showed increased water retention when LiCl was added to it [24]. The gel maintained 50% water retention after 7 days at 10% RH. In addition to salts, cross linkers or additives with hydroxyl groups such as glycols also increase the water retention of hydrogels. Ethylene glycol molecules in acrylamide/polyethyleneimine hydrogel inhibited crystallization and improved water retention due to the hydrogen bonding of hydroxyl groups [25]. Mxene nanoflakes incorporated hydrogel is also reported to show stable water retention properties, as Mxene acts as the cross linker and forms hydrogen bonds [26]. Also, hydrogel covered with a silicone elastomer has maintained water uptake for 2 days at 25 °C and 50% RH [27]. Another approach to preventing water evaporation from hydrogels is to add sugar [28]. When 40 wt.% of sucrose, one of the most commonly used disaccharides, was added to 1 wt.% agarose gel and dried at 50 °C, the gel retained moisture around 11 wt.%. A similar result was also observed when another disaccharide, trehalose, was added to agarose gel, showing moisture retention around 19 wt.% under the same gel composition and measurement conditions. One possible reason for the increased water retention capability of hydrogel from the added sugars is due to changes in the microstructure of the hydrogel. It has been reported that the pore size of hydrogels decreases when sugars are added, which could retard the evaporation of water [29].

Increasing the amount of ions in hydrogels by adding salts improves the ionic conductivity of hydrogels as well as the water retention. A poly(SBMA-co-AA) hydrogel doped with LiCl showed an ionic conductivity of 11 × 10^−2^ S/cm [24]. Polyvinyl alcohol doped with hydroxypropyl cellulose and NaCl showed an ionic conductivity of 3.4 × 10^−2^ S/cm [30]. Linear polymers with abundant C-O-C groups, such as poly(ethylene glycol) methyl ether acrylate (PEGMEA)/polyacrylamide, reduced the electrostatic potential of hydrated multivalent ions, thus significantly accelerating ion migration [31]. Co-polymerization of a hydrogel network with zwitterionic molecules also increased the ionic conductivity of the gel, which remained high even at a low temperature of −40 °C [32]. Cellulose-based hydrogel doped with bentonite and LiCl showed an ionic conductivity as high as 9 × 10^−2^ S/cm [33]. Although there has been a lot of effort to increase the water retention and the ionic conductivity of hydrogels, there has been little research on hydrogels that are composed entirely of biomass-derived eco-friendly polymers and additives.

This paper presents an ion-conductive agarose hydrogel with a high water-holding capacity, with additives of sucrose and sodium chloride (NaCl–Suc/A hydrogel). The structure of the hydrogel is schematically shown in Figure 1a. The NaCl provides ion current carriers to the agarose hydrogel, while the sucrose inhibits water evaporation from the hydrogel. First, how the NaCl and sucrose contents affect the time-dependent water loss and transmittance of the agarose hydrogels is discussed. The ionic conductivity of the NaCl–Suc/A hydrogel is calculated from its Nyquist plot and compared with those of the pristine agarose hydrogel and the agarose gel without sucrose. The changes in mechanical properties are confirmed through tensile testing. Additionally, the self-healing properties in terms of the ionic conductivity of the NaCl–Suc/A hydrogel are examined by repeated disconnecting and reconnecting of the gel. Finally, a pressure sensor is demonstrated using NaCl–Suc/A hydrogels as ionic conductors.

## 2. Materials and Methods

### 2.1. Materials

Agarose (BioReagent) and sucrose (≥99.5%, GC) were purchased from Sigma-Aldrich, St. Louis, MO, USA. NaCl (≥99%, extra pure) was purchased from Samchun Chemical Co., Ltd., Seoul, Republic of Korea. The monomer and the curing agent of polydimethylsiloxane (PDMS; Sylgard 184) were purchased from Dow Corning Co., Ltd., Midland, MI, USA.

### 2.2. Preparation of the NaCl–Suc/A Hydrogel

First, 1.0–5.0 g of sucrose was added to a vial filled with 4.8–8.8 mL of deionized water and bath-sonicated for 10 min to obtain different concentrations of sucrose solutions (10–50 wt.%). When the bath-sonication was complete, NaCl was added to the solutions at a concentration of 2 M. After that, 0.2 g of agarose was dissolved in the solutions at 95 °C to produce NaCl–Suc/A solutions. During heating, the solution was stirred at 500 rpm using a magnetic bar. Finally, the NaCl–Suc/A solution was poured into a PDMS mold and cooled to room temperature for gelation. NaCl–Suc/A hydrogel was obtained by removing it from the PDMS mold. As a control, pristine 2 wt.% agarose gel without additives was prepared using the same process.

### 2.3. Characterizations

The optical properties of the NaCl–Suc/A hydrogels were analyzed using ultraviolet–visible (UV–Vis) spectroscopy (Shimadzu, 1900i, Kyoto, Japan). The electrochemical impedance spectroscopy (EIS) measurement was performed using a potentiostat (BioLogic, SP-200, Seyssinet-Pariset, France) for gels with a size of 8 mm × 8 mm and 500 μm thick in an alternative current (AC) of 0.2 V in the frequency range of 0.1 Hz to 1000 kHz. The mechanical properties of the gels were measured using a universal testing machine (Instron, E3000LT, Norwood, MA, USA) at the speed of 1 mm/min with gel specimens made according to ASTM D638 [34] type 4. The resistance change of the gels was measured using a source-measure unit (Keithley, SMU 2450, Cleveland, OH, USA). The capacitance of the pressure sensors was measured and recorded using an LCR meter (Keysight, E4980AL, Santa Rosa, CA, USA).

### 2.4. Water Retention and Self-Healing Property Test

To investigate the water retention capability of the gels, agarose hydrogels with a size of 1 × 1 cm and 2 mm thick were prepared. The initial weight of the prepared hydrogels was measured, then the gels were dried in an oven set to 60 °C for 4 h. The water retention was calculated by comparing the water uptake before and after drying. While drying, the weight loss of the gels was measured every 10 min.

To evaluate the self-healing property of the NaCl–Suc/A hydrogel in terms of ionic conductivity, a piece of the gel with a size of 1 cm × 3 cm and 2 mm thick was cut into two pieces using a razor blade. The resistance through the gel was monitored while disconnecting and reconnecting the two pieces. To investigate the effect of the water retention capability of the gel on the self-healing ability in terms of ionic conductivity, the same self-healing test of ionic conductivity was performed on the cut pieces of NaCl–Suc/A hydrogel after drying them in the oven set to 60 °C for 4 h. All procedures were repeated for the control NaCl/A hydrogel for comparison.

### 2.5. Fabrication of Pressure Sensor Using NaCl–Suc/A Hydrogel

For the capacitive pressure sensor, the NaCl–Suc/A hydrogel was used to make thin film electrodes. The gel electrodes were fabricated by filling a square frame of PDMS (1 × 1 cm of inner space and a thickness of 1 mm, with outer dimensions of 1.5 × 1.5 cm) with the NaCl–Suc/A solution and then a cooling process was carried out for gelation. The resulting gel electrodes had dimensions of 1 × 1 cm and a thickness of 1 mm. The dielectric layer of the capacitive sensor was a thin film of PDMS (1.5 × 1.5 cm and 300 μm thick) sandwiched between the gel electrodes. The capacitive pressure sensor was encapsulated with additional PDMS layers (1.5 × 1.5 cm and 1 mm thick) at the top and the bottom. Two copper wires were inserted into the NaCl–Suc/A hydrogel layers for electrical connection. Pressure was applied to the sensor by placing 190 g or 350 g weight on the top. The capacitance changes of the sensor depending on the applied pressure were measured by applying an AC of 0.5 V and 300 kHz with an LCR meter.

## 3. Results and Discussion

### 3.1. Physical Appearance, Transmittance, and Ionic Conductivity of Agarose Gels with Additives

To investigate the effect of addition of sucrose and NaCl on the properties of the agarose hydrogel, four types of gels were prepared: pristine agarose gel, agarose gel with 2 M NaCl, agarose gel with 50 wt.% of sucrose, and agarose gel with 2 M NaCl and 50 wt.% of sucrose. The values of 2 M NaCl and 50 wt.% sucrose correspond to near-saturated concentrations of the solutes in water. Importantly, the high solubility of sugars in water allowed for stable gelation even with substantial amounts added. Figure 1b compares the appearance of the four gels. Notably, the gels containing sucrose are more transparent than the pristine agarose gel and the agarose gel with NaCl. The appearance corresponds to the UV–Vis spectra presented in Figure 1c. In the visible wavelength range (380–780 nm), the agarose gels doped with sucrose show much higher transmittance than those without sucrose. Disaccharide sucrose penetrates the crystalline fibers of polysaccharide agarose well, thereby forming hydrogen bonds [30]. Consequently, the agarose polymer network becomes more amorphous, making the gel more transparent. However, adding NaCl to the agarose gel decreases its transmittance. The high concentration of Na^+^ and Cl^−^ ions from the dissociation of NaCl salt in water may scatter the light. However, the NaCl–Suc/A hydrogel still has a high transmittance of over 95% in the wavelength region of >500 nm.

Adding NaCl can significantly increase the ionic conductivity of agarose hydrogels. To investigate the increase in ionic conductivity with the addition of NaCl, EIS measurement was performed (Figure 2). In the Nyquist plots, the *x*-intercept on the real axis represents the bulk resistance of the hydrogel. While the pristine agarose gel shows high resistance, adding NaCl salts decreases the resistance of the gel by >400 times. The conductivities were calculated using the following equation:(1)σ=l/(R×w×d),
where *R* is the measured resistance, *l* is the length, *w* is the width, and *d* is the thickness of the gels, respectively. Consequently, the ionic conductivity of the NaCl/A hydrogel is 5.1 × 10^−2^ S/cm. When sucrose is added to the NaCl/A hydrogel, the *x*-intercept on the real axis shifts to the right to a degree nearly proportional to the sucrose contents (Figure 2b and Appendix A). This indicates that sucrose lowers the ionic conductivity of NaCl–Suc/A hydrogel, probably because of the decreased ion mobility upon sucrose addition. The resulting ionic conductivity of the gel was found to be 1.6 × 10^−2^ S/cm when the concentration of sucrose was 50 wt.%. Nevertheless, even with the inclusion of 50 wt.% sucrose, the gel maintains a considerable level of ionic conductivity.

### 3.2. Water Retention of NaCl–Suc/A Hydrogels

To determine how effectively sucrose can inhibit water evaporation from ion-conductive hydrogels, the time-dependent water retention of hydrogels with different additives is compared under accelerated drying in an oven at 60 °C (Figure 3a,b). After 4 h, the pristine agarose gel loses most of its moisture and the NaCl/A hydrogel retains approximately 25% of its moisture. However, the Suc/A and NaCl–Suc/A hydrogels show water retention of ~40% and ~60%, respectively, after 4 h in the drying conditions. Thus, sucrose is more effective in improving the water retention of the agarose gel, while NaCl also contributes to suppressing water evaporation. Consequently, the water retention ability is further enhanced when the two solutes coexist.

The water evaporation rates of the agarose hydrogels are different upon addition of the sucrose and NaCl to the hydrogels. For all samples, it is generally observed that the rate of water loss gradually decreases. This is because, as the drying progresses from the surface of the hydrogel, it becomes more challenging for the water molecules to move to the surface and evaporate from the gel. The NaCl/A gel shows almost the same dehydration rate as the pristine agarose gel, but the rate decreased discontinuously after 120 min. This is probably due to precipitation of salt on the surface as the gel dries and the amount of water decreases [35,36]. After drying for 4 h, the appearance of the agarose gels becomes noticeably different (Figure 3b). The precipitation of NaCl salt is visually observed on the surface of the gels. This surface layer of precipitated salt could prevent water molecules from evaporating. In contrast, the agarose gel samples with sucrose, i.e., Suc/A and NaCl–Suc/A gels, maintain a semitransparent appearance without considerable surface precipitation of salt because of the high water retention (Figure 3b). Moreover, the hydroxyl groups of sucrose and ionized NaCl interact electrically, which also contributes to preventing salt precipitation [37,38,39]. Compared to the Suc/A hydrogel, the NaCl–Suc/A hydrogel has enhanced water retention capability. This is because the Na^+^ and Cl^−^ ions from the NaCl salt lower the water vapor pressure, which is a colligative property observed in solutions.

To investigate the effect of sucrose content on the water retention capability of the ion-conductive gels, the time-dependent water retention of NaCl–Suc/A gels with various amount of sucrose was measured (Figure 3c). As the sucrose content increases to its near-saturated concentration of 50 wt.%, not only the optical transparency of the NaCl–Suc/A gels increases, as shown in Appendix A, but also the water retention capability of the NaCl–Suc/A gels increases nearly proportionally, and the amount of NaCl salt precipitated on the gel surface decreases (Figure 3d). In this study, a sucrose concentration of 50 wt.% was used to minimize water loss.

### 3.3. Mechanical Properties of NaCl–Suc/A Hydrogels

Sucrose in the agarose gel matrix also influences the mechanical properties of the gel. Figure 4a shows the strain–stress curves of the agarose gels with different additives using a universal tensile testing machine in the uniaxial tension mode. When tensile strain is applied to the agarose gels, the gels show different modulus and yield stresses depending on the types of additives. Figure 4b shows the calculated Young’s modulus values. The elongation at break values of the gels with and without sucrose is 48% and 30%, respectively. The two types of gels containing sucrose show a noticeable increase in Young’s modulus and elongation at break compared to the pristine agarose gel. Conversely, the agarose gel doped with only NaCl shows a relatively insignificant difference from the pristine agarose gel. The Young’s modulus of the NaCl–Suc/A hydrogel is approximately 13% smaller than that of the Suc/A hydrogel.

The increased mechanical strength of the gel doped with sucrose implies two possible explanations. One explanation is structural change in the gels. When sucrose is added into the agarose gel, the degree of aggregation of the agarose polymer chains decreases [29]. This could lead to an increase in the cross-linking density in the entire agarose gel network, resulting in the increase in Young’s modulus and strain limit. The other is that the hydrogen bonding between sucrose and agarose enhances the tensile strength of the agarose gels. That is, the interaction between the agarose and sucrose increases the mechanical strength of the gel as well as its water retention capability. The decreased mechanical strength of the NaCl–Suc/A hydrogel compared to the Suc/A hydrogel could be due to the electrostatic interaction between the hydroxyl groups of sucrose and Na^+^ and Cl^−^ ions. The ions from the salt could weaken the hydrogen bonding between the hydroxyl groups of sucrose and agarose, reducing the mechanical strength of the entire polymer network.

### 3.4. Self-Healing Ionic Gel Conductors and Pressure Sensor

The hydrogel-based ionic conductor could have a self-healing property in terms of ionic conductivity because it is operated by ions in a liquid-like gel medium. Figure 5 shows experimental results to demonstrate the self-healing property of the ion-conductive hydrogel. A piece of the NaCl–Suc/A hydrogel was cut into two halves and attached again (Figure 5a). Then, the attached gels were left undisturbed at room temperature. After 10 min, the NaCl–Suc/A hydrogel forms a reasonably stable adhesion, enough to withstand pressure from human fingers. This self-healing ability of the gel restores the connection not only physically but also electrically (Figure 5b). When the gel was cut into two pieces and separated, the current decreased to zero, indicating that it was electrically disconnected. When the two pieces of gel were re-contacted after 10 s, the electrical connection was restored, demonstrating almost the same current value as before the cut. Furthermore, after the NaCl–Suc/A hydrogel that was cut into two pieces was stored in the oven at 60 °C for 4 h, the gel retained most of its self-healing property in relation to the electrical connection (Figure 5c). The NaCl/A hydrogel without sucrose also showed a similar self-healing property in terms of ionic conductivity. The higher current of the NaCl/A hydrogel compared to the NaCl–Suc/A hydrogel can be attributed to the slightly higher ion conductivity of the gel when sucrose is absent, as shown in Figure 2 and Appendix A. However, after drying in the oven for 4 h, the gel did not restore the electrical connection when reattached, due to the dehydration of the gels.

The NaCl–Suc/A hydrogel could be an efficient ionic conductor with water retention capability and high optical transparency. As an application example of the gel, a transparent capacitive pressure sensor was fabricated by sandwiching the PDMS dielectric layer between two NaCl–Suc/A hydrogel electrodes (Figure 6a). The fabricated pressure sensor is transparent and flexible. Both PDMS and agarose hydrogel have positive Poisson’s ratios close to 5 and expand horizontally against axial compression [40,41]. Therefore, upon pressure application, the area of the NaCl–Suc/A hydrogel electrode increases, increasing its capacitance. Figure 6b shows the capacitance change in response to the application of two magnitudes of pressure. When pressure is applied to the sensor, the capacitance of the device instantly increases. The change in capacitance is approximately proportional to the magnitude of the pressure, showing a reliable capacitance response to repeated applications of pressure.

## 4. Conclusions

This paper presents hydrogel-ionic conductors based on agarose gel doped with sucrose and NaCl salt. NaCl enables the gel with ion conductivity, and sucrose inhibits the dehydration of the gel while improving its optical transparency and mechanical properties. The hydrogel-based ion conductor exhibits a self-healing ability whereby the electrical conductivity that is lost when the gel is cut is restored by simply reconnecting the gel. Finally, a transparent pressure sensor based on the hydrogel-ionic conductor is demonstrated. The sensor can instantly detect the magnitude of the applied pressure by monitoring its capacitance change. Ionic conductors are expected to resolve issues in relation to hydrogels, such as drying out and low mechanical strength, and could be applied to various hydrogel ionics, such as wearable/implantable devices and sensors [42,43,44,45,46], soft robotics [47,48], and energy devices [18,49]. In particular, the gels discussed in this paper possess both high ionic conductivity and optical transparency, allowing them to be utilized as transparent electrodes and electrolytes in applications such as displays, solar cells, and optical sensors where light transmission is critical [50,51]. Additionally, the gel, which is composed of naturally derived agarose polymer, sucrose, and salt, is expected to be highly biocompatible, which would allow it to be actively utilized in applications that require human interfaces. For future work, investigating the effects of different types of salts and hydrophilic molecules such as saccharides on the conductivity and water retention behavior of agarose hydrogel would provide a more in-depth understanding of the relation between agarose hydrogel and the additives, thereby enabling a wide range of applications.

## Figures and Tables

**Figure 1 materials-17-00288-f001:**
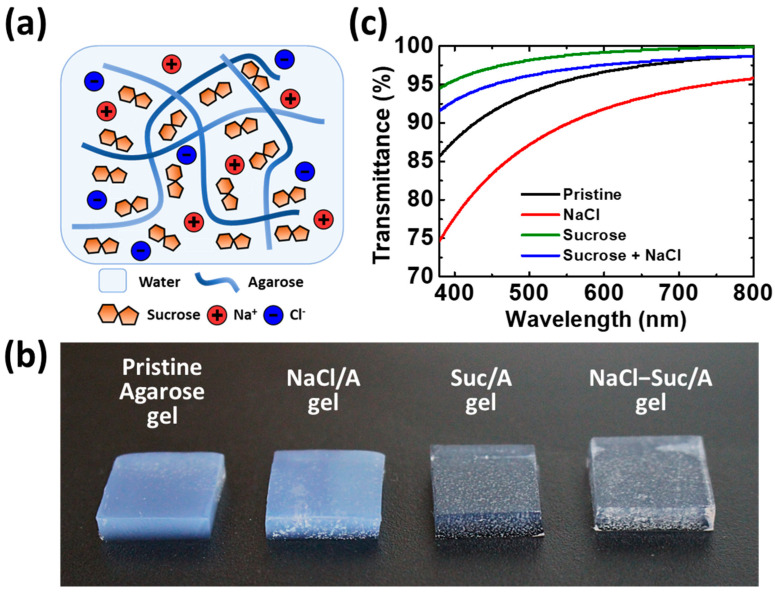
(**a**) Schematic diagram for the NaCl–Suc/A hydrogel; (**b**) photo images; and (**c**) UV–Vis transmittance spectra of the four agarose gels with different additives. The size of each gel is 1 × 1 cm^2^ with a thickness of 2 mm. The concentrations of NaCl and sucrose are 2 M and 50 wt.%, respectively.

**Figure 2 materials-17-00288-f002:**
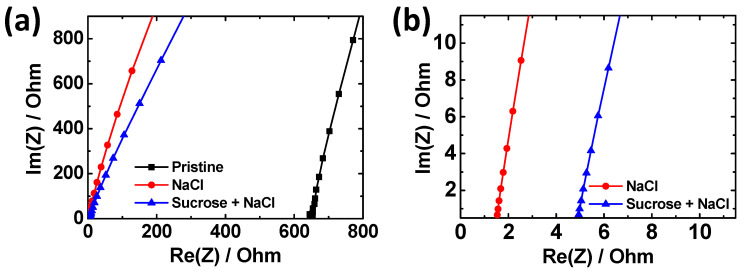
(**a**) Nyquist plots in the AC frequency range of 0.1 Hz–1000 kHz for hydrogels with different additives, while (**b**) shows the rescaled plots of (**a**). The concentrations of NaCl and sucrose are 2 M and 50 wt.%, respectively.

**Figure 3 materials-17-00288-f003:**
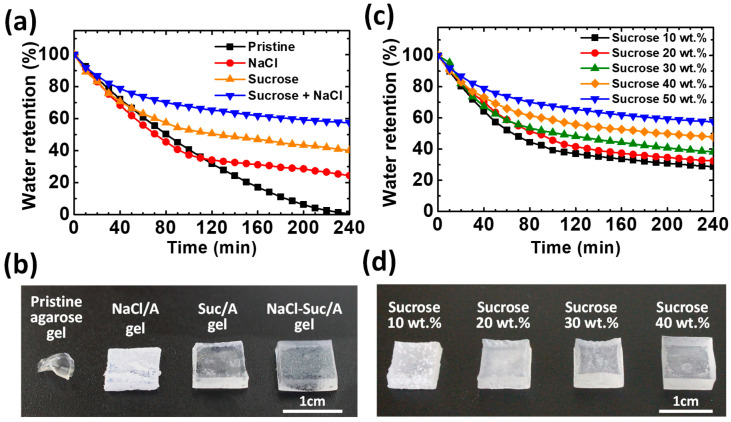
(**a**) Time-dependent water retention of the agarose gels with different additives in the oven at 60 °C. The concentrations of NaCl and sucrose are 2 M and 50 wt.%, respectively. (**b**) Photograph of the four gels in (**a**) after 4 h of drying at 60 °C. (**c**) Time-dependent water retention of the NaCl–Suc/A gels with different sucrose contents at 60 °C. All samples had 2 M NaCl added to them. (**d**) Photograph of four NaCl–Suc/A hydrogels with different sucrose weight ratios, after 4 h of drying at 60 °C.

**Figure 4 materials-17-00288-f004:**
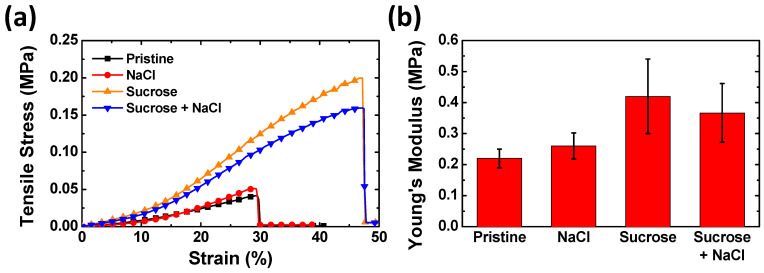
(**a**) Strain–stress curves and (**b**) Young’s modulus of the agarose hydrogels with different additives. The concentrations of NaCl and sucrose are 2 M and 50 wt.%, respectively.

**Figure 5 materials-17-00288-f005:**
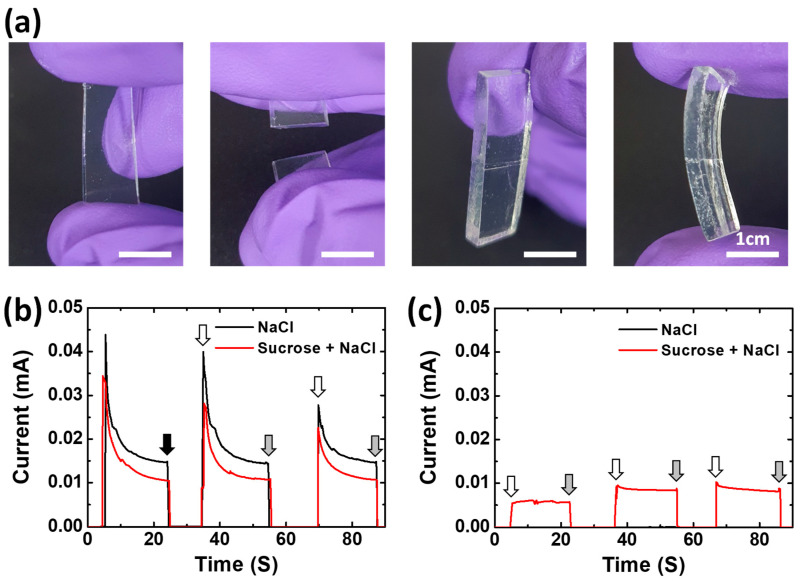
(**a**) Photographs showing the mechanical self-healing property of NaCl–Suc/A hydrogel. (**b**,**c**) Current changes in response to the disconnection and reconnection of ion-conductive hydrogels, (**b**) as prepared, and (**c**) after 4 h in the oven at 60 °C. The black, white, and gray arrows indicate when the gel was cut, reconnected, and detached again, respectively. In (**b**), the current measurement of the gel started at 5 s.

**Figure 6 materials-17-00288-f006:**
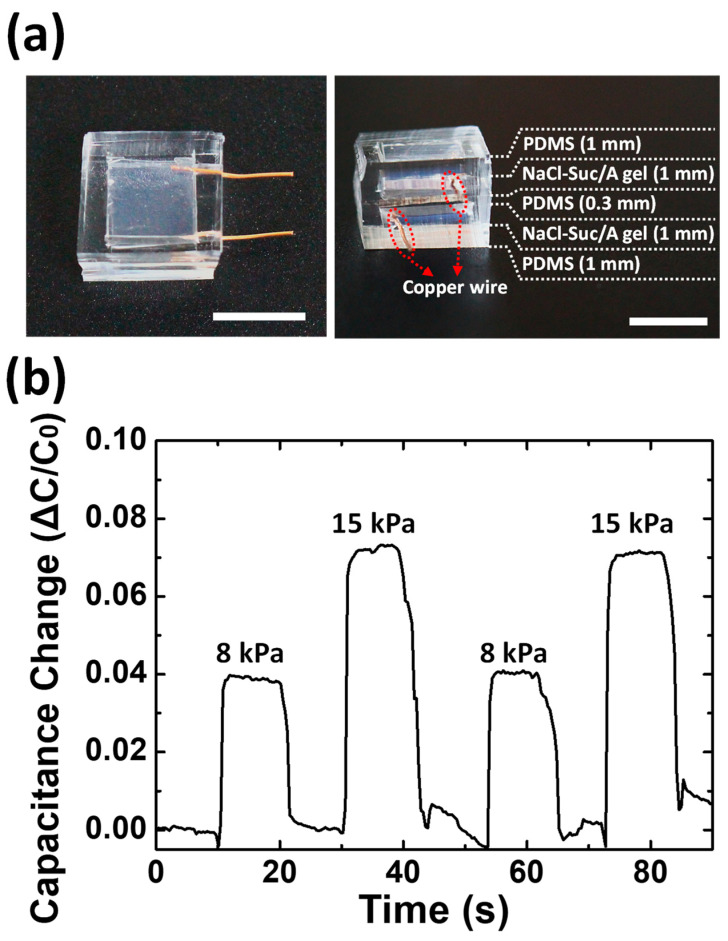
(**a**) A photograph of the prototype of the capacitive pressure sensor based on NaCl–Suc/A hydrogel electrodes. Scale bar = 1 cm. The gel layers are connected to the copper wires at both ends. (**b**) Capacitance response of the sensor to repetitive application of pressure with two magnitudes.

## Data Availability

The data in this study are available on request from the corresponding authors.

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
