# Peer review of "A Transparent Hydrogel-Ionic Conductor with High Water Retention and Self-Healing Ability"

_materials, 2024, doi:10.3390/ma17020288_

Round 1
Reviewer 1 Report
Comments and Suggestions for Authors
This paper presents hydrogel-ionic conductors that maintain high water content over time. The hydrogel comprises agarose polymer doped with sucrose and sodium chloride salt (NaCl–Suc/A hydrogel). The approach is correct and the results are well presented. I personally give minor revision.
1. The author mentioned the self-healing property of hydrogels in the manuscript. Therefore, I suggest that the author could investigate the self-healing ability of the hydrogel. Specific experimental procedures can be referred to in the literature: DOI: 10.1016/j.jconrel.2023.01.049.
2. The Young's modulus of the hydrogel in Figure 4b is discussed. I also recommend that the author increase the number of samples tested for each group.
3. The analysis of the overall mechanical properties of the hydrogel should be more in-depth.
Comments on the Quality of English LanguageMinor editing of English language required.
Author Response
Please find attached the response file. We appreciate the reviewer's time and effort.

Reviewer 2 Report
Comments and Suggestions for Authors
The submitted manuscript provides for the preparation and evaluation of sucrose and sodium chloride loaded agarose gels. Various studies have been undertaken on the prepared hydrogels including UV spectrum scanning, electrochemical impedance spectroscopy, mechanical strength testing, self-healing testing, and the fabrication of a flexible pressure sensor. The study is interesting; however, the authors have not adequately addressed the novelty of this study and its importance, especially with regards to the new material that has been prepared, which is essential for this journal, and the rationale for the development of the flexible pressure sensor. A list of further comments/corrections has been provided below.
1. Abstract: The abstract needs to be rewritten and should focus on the novelty and potential impact of the undertaken research. Additionally, the flexible pressure sensor is only briefly mentioned and its relevance to the research study is not provided.
2. Introduction: Agarose, sucrose and NaCl are commonly used molecules and have been extensively researched previously. The authors have provided a good introduction on the benefits of adding ionic agents to hydrogels, however no introduction is provided to the use of sucrose for strengthening of the hydrogel, on self-healing hydrogels or on the developed flexible pressure sensor.
3. Hydrogel Characterizations: the methods need to be expanded to include the controls used in the study and the comparison to the pristine agarose hydrogel.
4. The characterization methodology states that the morphology of the hydrogels has been evaluated using an optical microscope. The results of this have not been presented in the manuscript.
5. Water retention and self-healing property test: The length of time used for the drying of the hydrogel should be provided. Additionally, the self-healing property test should also be expanded to include the conditions and timing of the tests. Also, was any further imaging and structural characterization undertaken to validate the the self-healing potential of the prepared hydrogels?
6. Fabrication of pressure sensor using NaCl–Suc/A hydrogel: the weights used for this test should be provided as well as any controls used in the analysis.
7. Figure 3: Images of the hydrated hydrogels prior to drying should be provided.
8. Figure 5: the results of in these images show that the addition of sucrose decreases the electrical conduction of the gels. This should be discussed in the manuscript.
9. The authors should provide any potential applications for the developed hydrogels based on the results seen in this study.
10. The supplementary data has not been mentioned in the main text and should be considered for inclusion as it reports on the different NaCl-Suc/A hydrogels prepared.
Comments on the Quality of English LanguageMinor language errors are present in the manuscript. A proof-read of the complete manuscript is advised.
Author Response

(The authors gave the same response as above.)

Reviewer 3 Report
Comments and Suggestions for Authors
In this paper, the authors examine the properties of ion-conducting agarose hydrogels with high water retention capacity supplemented with sucrose (Suc) and sodium chloride. They discuss how the NaCl and sucrose content affects the time-dependent water loss and permeability of the agarose hydrogels, but only at specific ratios. Although only pictures of varying concentrations of both NaCl and Suc are included in the supporting information, these are additives that have independent effects, since originally adding Suc would increase permeability and adding NaCl would increase turbidity. Therefore, it is believed that this system can determine the optimal respective concentrations of Suc and NaCl that give maximum ionic conductivity, strongest tensile strength, and excellent self-healing properties by independently varying the amount of Suc and NaCl added. Since this manuscript lacks such basic information, it may be essential to provide additional information on them. This reviewer recommends that this manuscript be published in this journal with these additional information.
Author Response

(The authors gave the same response as above.)

Round 2
Reviewer 2 Report
Comments and Suggestions for Authors
The authors have substantially improved on the initial manuscript submission and have adequately addressed all comments provided. Minor editing of the revised text is advised before this manuscript can be accepted for publication. A list of amendments to be made has been provided below.
1. Abstract, Line 21. 'property' should be 'properties'.
2. Lines 39-41: the added text 'are generally have self-healing property' should be reworded.
3. Lines 76-78: 'when sugars added' should be 'when sugars are added'.
4. Line 152: the added text 'process was undergo' should be reworded.
Comments on the Quality of English LanguageThere are minor language errors that the authors should address before this manuscript can be accepted for publication. A list of language errors has been provided in the 'Comments and Suggestions for Authors'.
Author Response
We are thankful to the reviewer for taking a close look at our manuscript and pointing out the errors. We have fixed the errors pointed out in the revised manuscript and highlighted them in green.